# Membrane insertion of α-xenorhabdolysin in near-atomic detail

Evelyn Schubert[1], Ingrid R Vetter[2], Daniel Prumbaum[1], Pawel A Penczek[3], Stefan Raunser[1]*

[1]Department of Structural Biochemistry, Max Planck Institute of Molecular Physiology, Dortmund, Germany; [2]Department of Mechanistic Cell Biology, Max Planck Institute of Molecular Physiology, Dortmund, Germany; [3]Department of Biochemistry and Molecular Biology, Houston Medical School, The University of Texas, Houston, United States

**Abstract** α-Xenorhabdolysins (Xax) are α-pore-forming toxins (α-PFT) that form 1–1.3 MDa large pore complexes to perforate the host cell membrane. PFTs are used by a variety of bacterial pathogens to attack host cells. Due to the lack of structural information, the molecular mechanism of action of Xax toxins is poorly understood. Here, we report the cryo-EM structure of the XaxAB pore complex from *Xenorhabdus nematophila* and the crystal structures of the soluble monomers of XaxA and XaxB. The structures reveal that XaxA and XaxB are built similarly and appear as heterodimers in the 12–15 subunits containing pore, classifying XaxAB as bi-component α-PFT. Major conformational changes in XaxB, including the swinging out of an amphipathic helix are responsible for membrane insertion. XaxA acts as an activator and stabilizer for XaxB that forms the actual transmembrane pore. Based on our results, we propose a novel structural model for the mechanism of Xax intoxication.

DOI: https://doi.org/10.7554/eLife.38017.001

**\*For correspondence:**
stefan.raunser@mpi-dortmund.mpg.de

**Competing interests:** The authors declare that no competing interests exist.

## Introduction

Pore-forming toxins (PFTs) are soluble proteins produced by bacteria and higher eukaryotes, that spontaneously form pores in biomembranes and act as toxins (*Dal Peraro and van der Goot, 2016*). Dependent on their transmembrane region, which is formed either by α-helices or β-strands, PFTs are classified as α-PFTs and β-PFTs (*Iacovache et al., 2008*; *Bischofberger et al., 2012*). A common trait of all PFTs is the conversion from a soluble monomer to a membrane-embedded oligomer (*Bischofberger et al., 2012*); however, a different mechanism has been recently found for ABC toxins (*Gatsogiannis et al., 2016*). Specific targeting of the PFTs to the host membrane involves mostly recognition of specific proteins, glycans or lipids on the target membrane. Conformational changes resulting in the oligomerization and membrane perforation are triggered by receptor binding, catalytic cleavage, pH change or other factors (*Iacovache et al., 2008*). The sequential order of oligomerization and membrane penetration including the formation of an oligomeric pre-pore is still a matter of debate (*Cosentino et al., 2016*). The size of the oligomers ranges from tetrameric pores in Cry1Aa (*Gómez et al., 2014*) and heptameric pores in the anthrax protective antigen (*Jiang et al., 2015*) to 30–50-meric pores in cholesterol-dependent cytolysins (CDCs) (*Dal Peraro and van der Goot, 2016*; *Hotze and Tweten, 2012*).

PFTs can be further divided into two groups (*Iacovache et al., 2010*). PFTs of the first group perforate membranes by forming stable pores resulting in an uncontrolled influx and efflux of ions and other biomolecules. This destroys ion gradients and electrochemical gradients at the membrane. The toxins of the second group also perforate the membrane, but use the transmembrane channel to specifically translocate toxic enzymes into the host. Binary toxins, also called AB toxins

**eLife digest** Some bacteria make toxins that punch large holes into the membranes of host cells, destroying them like a puncture destroys a football. These "pore-forming toxins" allow many bacterial species to infect a variety of organisms, from insects to humans. Some sophisticated pore-forming toxins, such as the anthrax toxin, do not only form a pore but also use it to flood lethal toxins into the cell to kill it.

One bacterium called *Xenorhabdus nematophila* punctures the membranes of insect cells, using the same type of pore-forming toxins that other bacteria use to infect humans. Previous research has shown that two proteins – components A and B – form these pore-forming toxins. Given this two-protein formation, some scientists predicted these pore-forming toxins might act like those of the anthrax bacterium: one component forms the pore; the other component poisons the cell. But without detailed images of this pore-forming toxin's structure, understanding exactly how these two components work together is almost impossible.

To explore how components A and B operate within *X. nematophila,* Schubert et al. captured images of the molecular structure of the two proteins. Common methods reliant on X-rays and electron microscopes revealed the layouts of both components. By visualizing the proteins at different stages, Schubert et al. observed key structural changes that enable them to form the pore and puncture a host cell.

Component A binds to component B's back, forming a subunit – twelve to fifteen of which then conjoin as the pore-forming toxin. Schubert et al. conclude that component A stabilizes each subunit on the membrane and activates component B, which then punctures the membrane by swinging out its lower end. Unlike the anthrax pore-forming toxin, both components collaborate to form the pore complex and puncture the membrane. These results provide a foundation of knowledge about what these toxins look like and how they operate. More research building upon this structural analysis may help scientists develop antibiotics that prevent bacteria from destroying human cells.
DOI: https://doi.org/10.7554/eLife.38017.002

(*Odumosu et al., 2010*) and also recently characterized ABC toxins (*Meusch et al., 2014*) belong to the latter group. A prominent AB toxin is the anthrax toxin (*Collier and Young, 2003*), where component B, the protective antigen, forms a translocation pore through which lethal factor or edema factor, different A components, are translocated.

The members of α-PFTs show a high structural diversity. They include proteins mainly consisting of α-helical structures (bax, colicins) or β-strand motifs with a single helix responsible for membrane insertion (actinoporins) (*Dal Peraro and van der Goot, 2016*; *Parker and Feil, 2005*). Their transmembrane regions are all composed of hydrophobic or amphipathic regions buried within the core structure of the soluble monomer. Therefore, a conformational change that exposes the hydrophobic or amphipathic region is required for successful membrane insertion. The structures of cytolysin A (ClyA) (*Wallace et al., 2000*; *Mueller et al., 2009*) and fragaceatoxin C (FraC) (*Wallace et al., 2000*; *Mueller et al., 2009*) of both the monomer and oligomer gave the first structural insight into the mechanism of action of this class of PFTs.

In contrast to α-PFTs, the structures of many β-PFTs, such as members of the cholesterol-dependent cytolysins (*Hotze and Tweten, 2012*), hemolysin and aerolysin family (*Dal Peraro and van der Goot, 2016*), have been determined in their monomeric and pore conformation. The transmembrane β-strands in the soluble monomers of β-PFTs are typically amphipathic with small hydrophobic patches that upon oligomerization form a hydrophobic membrane-spanning β-barrel.

α-Xenorhabdolysin is a PFT that has been first isolated from the bacterium *Xenorhabdus nematophila* (*Ribeiro et al., 2003*). Xenorhabdolysins are also found in other entomopathogenic bacteria, such as *Photorhabdus luminescens*, and human pathogenic bacteria, such as *Yersinia enterocolitica* and *Proteus mirabilis* (*Vigneux et al., 2007*). They are composed of two subunits, namely XaxA (45 kDa) and XaxB (40 kDa) and are only active when the two components act together (*Vigneux et al., 2007*). Xenorhabdolysins, which were suggested to be binary toxins (*Vigneux et al., 2007*; *Zhang et al., 2014*), perforate the membranes of erythrocytes, insect granulocytes and phagocytes and induce apoptosis (*Vigneux et al., 2007*; *Zhang et al., 2014*). The mechanism of action of

xenorhabdolysins including the interaction between components A and B, oligomerization and pore formation has remained enigmatic so far.

Structural prediction using the PHYRE2 server (*Kelley and Sternberg, 2009*) does not yield any significant similarities for XaxB. XaxA cytotoxins, however, are predicted to be similar to two pore-forming cytolysins, Cry6Aa from *Bacillus thuringiensis* (*Huang et al., 2016*) and binding component B of hemolysin BL (Hbl-B) from *Bacillus cereus* (*Madegowda et al., 2008*). The best characterized cytolysin is probably ClyA from *Escherichia coli* and *Salmonella enterica* strains. The structure of ClyA has been determined in its soluble form (*Wallace et al., 2000*) and pore conformation (*Mueller et al., 2009*) and the mechanism of pore formation mechanism has been extensively studied (*Roderer and Glockshuber, 2017*). However, in contrast to XaxAB, ClyA only contains one component. Thus, despite the structural similarity, the mechanism of action must be different.

So far structural data on xenorhabdolysins are missing limiting our understanding of these important type of toxins. Here, we used a hybrid structural biology approach combining X-ray crystallography and electron cryomicroscopy (cryo-EM) to determine the crystal structures of XaxA and XaxB from *Xenorhabdus nematophila* as soluble monomers and the cryo-EM structure of the XaxAB pore complex.

## Results and discussion

### Structure of XaxA and XaxB soluble monomers

In two different experiments, we independently expressed and purified XaxA and XaxB (Materials and methods). The protein quantity and quality of both proteins was sufficient (*Figure 1—figure supplements 1a–b* and *2a–b*) to perform crystallization experiments. We obtained well diffracting crystals of both XaxA and XaxB in their soluble monomeric form and solved their structures to 2.5 and 3.4 Å, respectively (*Figure 1a–b*, *Table 1*).

Both XaxA and XaxB have a long rod-shaped structure and are mainly composed of α-helices (*Figure 1a–b*). XaxA and XaxB have a similar domain organization. They both contain a tail domain that is formed by a five-helix bundle (αA, αB, αC, αG and αH) and elongated neck and head domains. The five-helix bundle motif has so far only been described for ClyA and ClyA-type toxins (*Roderer and Glockshuber, 2017*). Like in the case of ClyA-type toxins the N-terminal helices (αA) of XaxA and XaxB are significantly shorter than in ClyA, where it plays a crucial role in pore formation (*Figure 1—figure supplement 4*). Interestingly, XaxA contains two large loops connecting the helices, a big hook-shaped loop (lp1, aa 136–169) between helices αB and αC at the top of the tail domain and an additional loop (lp2, aa 202–215) dividing helix αC (*Figure 1a*). The four XaxB molecules in the asymmetric unit differ considerably in their tail domain (*Figure 1—figure supplement 5*). Especially, helices αB and αC that protrude slightly from the five-helix-bundle take different positions. Although this might be due to tight crystal packing, it also indicates a certain degree of flexibility of the tail domain of XaxB.

A long coiled-coil structure, composed of a continuous helix (αG) and another one that is divided into three (XaxA: αC1, αC2 and αD) or two (XaxB: αC, αD) segments, form the backbone of XaxA and XaxB. It connects all domains and forms in both XaxA and XaxB the neck and head domain. The neck domain, that is approximately 35 Å in length, does not exist in ClyA-type toxins, which are in general more compact (*Figure 1—figure supplement 4a*). In XaxA, the tip of the coiled-coil, predicted as hydrophobic transmembrane region is not resolved in our crystal structure, however, secondary structure predictions for this region suggest a continuation of the coiled-coil (*Figure 1—figure supplement 6*).

In contrast to XaxA the head domain of XaxB contains in addition to the central coiled-coil a helix-loop-helix motif, dividing helix αE and a 21-residue long amphipathic helix (αF). The highly conserved hydrophobic face of helix αF is oriented toward helices αD and αG and thereby shielded from the solvent (*Figures 1b* and *2*). The conformation of the head domain is stabilized by conserved hydrophobic as well as electrostatic interactions, including putative salt bridges (*Figure 2*).

In general, the overall fold of the soluble monomers is similar to that of ClyA from *Escherichia coli* (*Wallace et al., 2000*) or ClyA-type toxins, such as Cry6Aa from *Bacillus thuringiensis* (*Huang et al., 2016*), non-hemolytic enterotoxin A (NheA) (*Ganash et al., 2013*), and binding component B of hemolysin BL (Hbl-B) from *Bacillus cereus* (*Madegowda et al., 2008*) (*Figure 1—figure supplement*

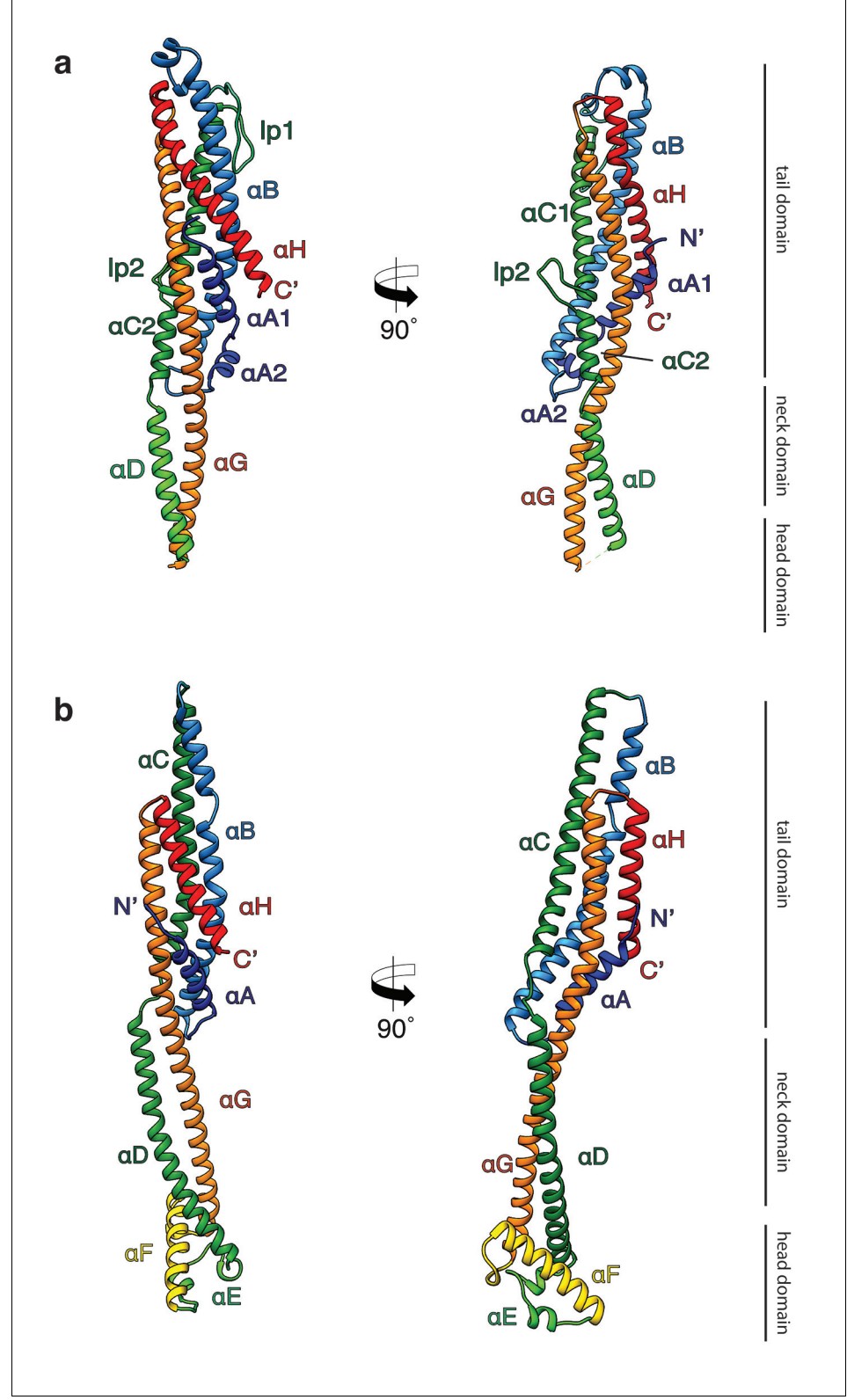

**Figure 1.** Crystal structures of XaxA and XaxB in their soluble monomeric form. (a) Ribbon representation of the atomic model of the XaxA soluble monomer. (b) Ribbon representation of the XaxB soluble monomer. Each helix is depicted in a different color and labeled accordingly.

DOI: https://doi.org/10.7554/eLife.38017.003

*Figure 1 continued on next page*

*Figure 1 continued*

The following figure supplements are available for figure 1:

**Figure supplement 1.** Purification and negative stain EM of XaxA.
DOI: https://doi.org/10.7554/eLife.38017.004
**Figure supplement 2.** Purification and negative stain EM of XaxB.
DOI: https://doi.org/10.7554/eLife.38017.005
**Figure supplement 3.** Analytical size exclusion chromatography and negative stain EM of XaxAB.
DOI: https://doi.org/10.7554/eLife.38017.006
**Figure supplement 4.** Comparison of XaxA and XaxB with ClyA-type toxins.
DOI: https://doi.org/10.7554/eLife.38017.007
**Figure supplement 5.** Superposition in stereo representation of the four XaxB molecules in the asymmetric unit.
DOI: https://doi.org/10.7554/eLife.38017.008
**Figure supplement 6.** Secondary structure prediction of XaxA generated by PSIPRED.
DOI: https://doi.org/10.7554/eLife.38017.009

*4a*). An important feature of ClyA and ClyA-type cytotoxins is the typical tongue motif that inserts into the membrane during pore formation (*Mueller et al., 2009*) (*Figure 1—figure supplement 4a, b*). In ClyA, Hbl-B, NheA, and Cry6Aa the tongue is formed by a hydrophobic or amphipathic β-hairpin or a large hydrophobic loop (*Wallace et al., 2000*; *Huang et al., 2016*; *Madegowda et al., 2008*). Interestingly, in the case of XaxB the tongue is formed by an amphipathic helix, while XaxA does not contain such motif. Comparing the structure of XaxA with that of its pore conformation (see below) suggests that XaxA is already in its extended conformation as soluble monomer.

It is tempting to speculate that the function of the N-terminal helix and β-tongue in ClyA has been evolutionary compensated in multicomponent toxins, such as XaxAB, NheABC and Hbl-ABC that only contain a short N-terminal helix. In the case of XaxAB, the hydrophobic helices of XaxA that enter the membrane in the pore, are functionally equivalent to the hydrophobic β-tongue of ClyA, The β-tongue likely inserts first into the membrane, where it rearranges into two α-helices (*Mueller et al., 2009*). Similar to XaxA, these helices only span half the membrane. αF and the helix-turn-helix motif αE of XaxB, that span the complete membrane in the XaxAB pore, would substitute the N-terminal membrane-spanning helix of ClyA.

## Structure of the XaxAB pore complex

To investigate the pore complex formed by XaxA and XaxB, we planned to induce pore formation in vitro and analyze the structure of the complex by single particle electron cryomicroscopy (cryo-EM). We first mixed both soluble monomers, incubated them with a variety of detergents and analyzed the pores by negative stain electron microscopy. We could indeed observe pore formation in most cases; however, the choice of detergent greatly influenced the size and homogeneity of the observed crown-shaped pore complexes. Some detergents induced the formation of star-like aggregates or differently sized pores (*Figure 3—figure supplement 1*). We observed the most homogenous distribution of XaxAB pore complexes, that appear as crown-shaped structures, after incubating the monomers with 0.1% Cymal-6 (*Figure 3—figure supplement 1c*, *Figure 3—figure supplement 2b*). The average diameter of the pores was ~250 Å. However, the pores had the tendency to aggregate and were not suitable for further structural investigations. Interestingly, when we incubated soluble monomers of XaxA and XaxB in the absence of detergents at room temperature, we observed the formation of higher oligomers but not of complete pores (*Figure 1—figure supplement 3a,b*). This is not the case when XaxA and XaxB are not mixed (*Figure 1—figure supplements 1* and *2*). This indicates that heterodimerization and oligomerization of XaxAB can happen independently of the hydrophobic environment provided by detergents or a lipid bilayer and may happen prior to pore formation also in vivo.

To improve the homogeneity of our XaxAB pore complexes, we exchanged Cymal-6 with amphipols and separated the amphipol-stabilized XaxAB pores from the aggregates by size exclusion chromatography (*Figure 3—figure supplement 2c,d*). The thus obtained pore complexes were homogeneous and suitable for single particle cryo-EM.

Analyzing the single particles by two-dimensional clustering and sorting in SPHIRE (*Yang et al., 2012*; *Moriya et al., 2017*) revealed populations of XaxAB pores with different numbers of subunits

**Table 1.** Data collection and refinement statistics.

| | | XaxA | XaxB |
|---|---|---|---|
| Data collection | | | |
| Wavelength (Å) | SLS | 2.07505 | 0.97793 |
| | PETRA | 1.8233 | |
| Resolution range (Å) | | 44.48–2.5 (2.589–2.5) | 48.15–3.4 (3.521–3.4) |
| Space group | | P 21 21 21 | P 21 21 21 |
| Cell dimensions a, b, c (Å) | | 67.27 90.83 153.03 | 88.7 99.41 194.15 |
| α, β, γ (°) | | 90 90 90 | 90 90 90 |
| Molecule no. in AU | | 2 | 4 |
| Total reflections | | 996,585 (92,922) | 961,813 (91,076) |
| Unique reflections | | 33,174 (3,258) | 24,297 (2,378) |
| Multiplicity | | 30.0 (28.5) | 39.6 (38.3) |
| Completeness (%) | | 99.91 (99.94) | 99.91 (99.96) |
| Mean I/σ(I) | | 25.11 (2.38) | 14.23 (0.82) |
| Wilson B-factor | | 58.45 | 137.29 |
| R-merge | | 0.1055 (1.722) | 0.2846 (6.285) |
| R-meas | | 0.1073 (1.753) | 0.2883 (6.369) |
| CC1/2 | | 1 (0.872) | 0.999 (0.493) |
| CC* | | 1 (0.965) | 1 (0.813) |
| Refinement | | | |
| Reflections used in refinement | | 33,167 (3,257) | 24,289 (2377) |
| Reflections used for R-free | | 1659 (173) | 1215 (119) |
| $R_{work}/R_{free}$ (%) | | 23.84/28.57 (35.19/42.56) | 26.38/30.52 (37.37/40.11) |
| CC(work)/CC(free) | | 0.958/0.943 (0.786/0.715) | 0.957/0.941 (0.613/0.442) |
| Average B-factor (Å$^2$) | | 77.47 | 142.42 |
| No. atoms in AU | | 5373 | 10,624 |
| Macromolecules | | 5348 | 10,624 |
| Solvent | | 249 | |
| Protein residues | | 678 | 1329 |
| r.m.s. deviations: | | | |
| RMS (bonds) | | 0.004 | 0.004 |
| RMS (angles) | | 0.87 | 0.72 |
| Ramachandran favored (%) | | 99.4 | 97.50 |
| Ramachandran allowed (%) | | 0.6 | 2.20 |
| Ramachandran outliers (%) | | 0.00 | 0.3 |
| Rotamer outliers (%) | | 1.32 | 3.82 |
| Clashscore | | 3.69 | 3.99 |

Values for the highest resolution shell are inside brackets.

*For XaxA multiple datasets were collected from one crystal at the PXIII-X06DA beamline at the Swiss Light Source and at the DESY PETRA III beamline P11.

DOI: https://doi.org/10.7554/eLife.38017.010

(*Figure 3—figure supplement 3*). Most of the complexes contain either 12, 13, 14 or 15 subunits. We separated the different populations by multi-reference alignment and solved the structure of the different complexes in SPHIRE (*Moriya et al., 2017*) (Materials and methods, *Figure 3—figure supplement 4–5*). The average resolutions of the reconstructions were 5, 4, 4.2 and 4.3 Å for do-, tri-, tetra-, and pentadecameric pores, respectively (*Figure 3—figure supplements 4–5*). We used the

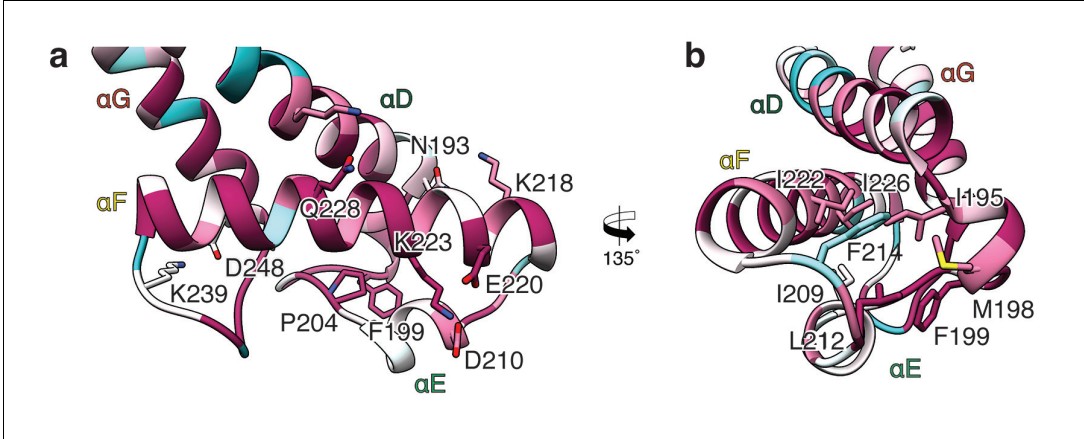

**Figure 2.** Interactions in the head domain of the XaxB monomer. (**a**) The head domain of XaxB is stabilized by hydrophobic and electrostatic interactions including putative salt bridges. (**b**) The hydrophobic face of the amphipathic helix αE is shielded in the soluble monomer by hydrophobic interactions with the rest of the head domain. Figures are colored by degree of conservation based on a sequence alignment of XaxB with homologous sequences from different bacterial species from 100% (magenta) to 0% (cyan).

DOI: https://doi.org/10.7554/eLife.38017.011

highest resolved cryo-EM density of the tridecameric pore complex to build an atomic model of XaxAB (*Video 1*, *Figure 3*, *Table 2*). The high quality of the map allowed both models to be almost completely built, except for the first residues of the N-terminal helix αA in XaxA (aa 1–40) and XaxB (aa 1–12). These regions are also not resolved in the crystal structures indicating a high flexibility of the N-termini.

The pore complexes have a total height of 160 Å and depending on the number of subunits a diameter of 210 to 255 Å. Each subunit consists of a XaxAB heterodimer with XaxA bound to the back of XaxB. This results in a localization of XaxA on the periphery of the pore, whereas XaxB resides more at the center of the complex lining the inner pore lumen (*Figure 3*, *Video 1*). Interestingly, the transmembrane helices of XaxA that fortify the inner ring of helices of XaxB, do not completely span the membrane (*Figure 3—figure supplement 6*). The arrangement of the components clearly shows that XaxAB is not a binary toxin as suggested (*Vigneux et al., 2007*; *Zhang et al., 2014*), but rather a bi-component toxin, such as BinAB from *Lysinibacillus sphaericus* (*Colletier et al., 2016*) and leukocidin A and B (LukGH and SF) from *Staphylococcus aureus* (*Badarau et al., 2015*) where both proteins contribute to building the pore.

Depending on the number of subunits, the inner diameter of the pore narrows down from 140 to 170 Å at the membrane-distal part to 40–55 Å at the transmembrane region. The inner surface of the pore is hydrophilic and mostly negatively charged suggesting a preference for positively charged ions and molecules (*Figure 3—figure supplement 7*). At the outside, the pore complex has a conserved highly hydrophobic band of 40 Å corresponding to the transmembrane region (*Figure 3—figure supplement 7*). The hydrophobic band merges into a positively charged stretch that is formed by the conserved arginine and lysine residues of XaxA (R290, K291, K293, K295, K301) (*Figure 3—*

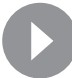

**Video 1.** Cryo-EM map of XaxAB in its pore state. Molecular model and cryo-EM map of the tridecameric XaxAB pore complex from *Xenorhabdus nematophila*, showing the overall structure of the pore complex. XaxA and XaxB subunits are colored in green and yellow, respectively.

DOI: https://doi.org/10.7554/eLife.38017.023

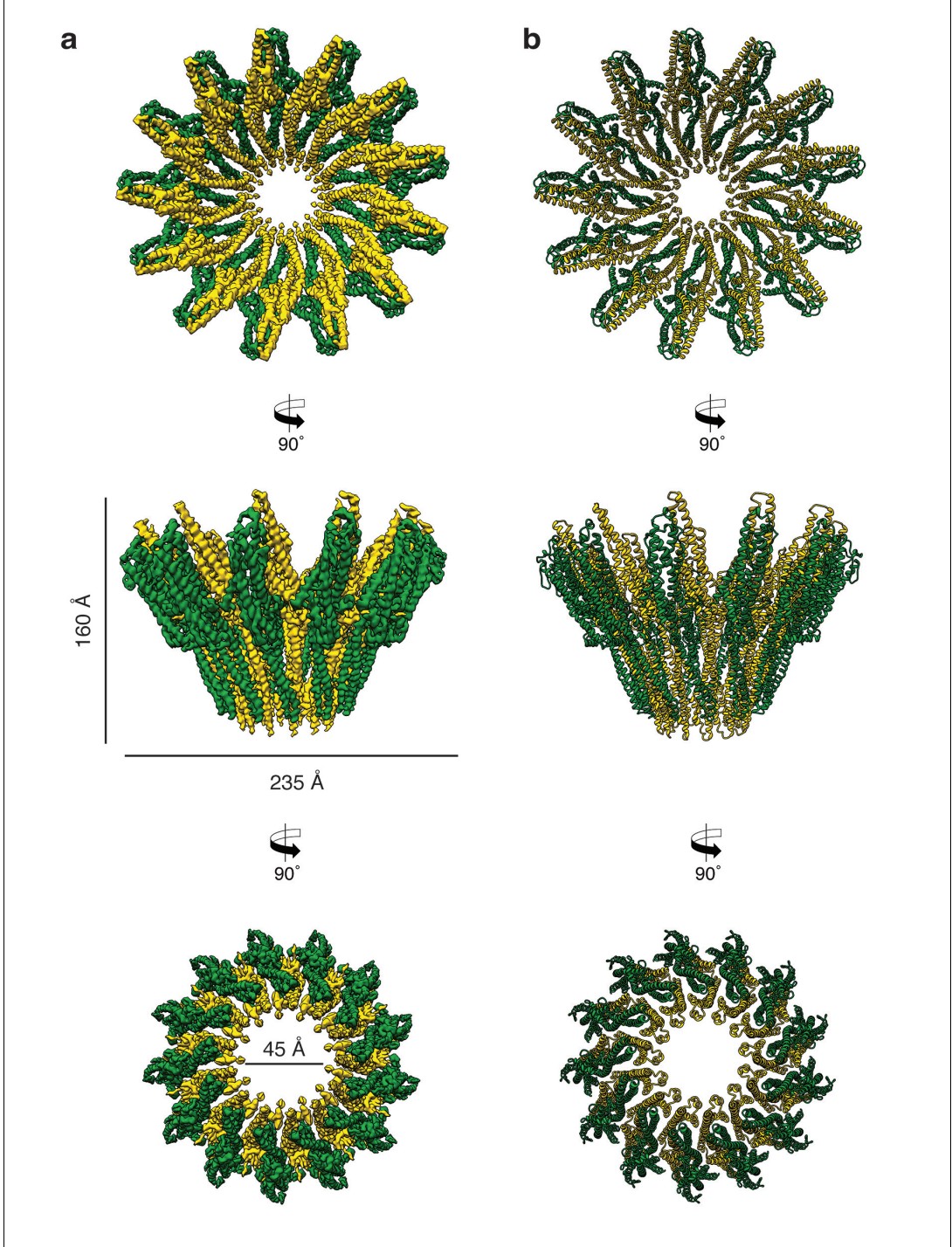

**Figure 3.** Cryo-EM structure of the tridecameric XaxAB pore complex. (a) Cryo-EM density map of tridecameric XaxAB pores shown as top, side and bottom view. XaxA and XaxB are colored in green and yellow, respectively. (b) Ribbon representation of the atomic model of XaxAB. Colors shown as in (a).

DOI: https://doi.org/10.7554/eLife.38017.012

The following figure supplements are available for figure 3:

**Figure supplement 1.** Pore formation of XaxAB induced by different detergents and analyzed by negative stain EM.
DOI: https://doi.org/10.7554/eLife.38017.013

**Figure supplement 2.** Analytical size exclusion chromatography and negative stain EM of XaxAB in Cymal-6 and amphipols.
DOI: https://doi.org/10.7554/eLife.38017.014

*Figure 3 continued*

**Figure supplement 3.** 2-D class averages of XaxAB pores with different numbers of subunits.
DOI: https://doi.org/10.7554/eLife.38017.015
**Figure supplement 4.** Single particle processing workflow of XaxAB structure determination.
DOI: https://doi.org/10.7554/eLife.38017.016
**Figure supplement 5.** Cryo-EM structure of tridecameric XaxAB.
DOI: https://doi.org/10.7554/eLife.38017.017
**Figure supplement 6.** Transmembrane domains of XaxAB embedded in amphipols.
DOI: https://doi.org/10.7554/eLife.38017.018
**Figure supplement 7.** Biophysical properties of the XaxAB pore.
DOI: https://doi.org/10.7554/eLife.38017.019
**Figure supplement 8.** Amino acid sequence alignment and conservation of the transmembrane region of XaxA (a) and XaxB (b).
DOI: https://doi.org/10.7554/eLife.38017.020
**Figure supplement 9.** Comparison of the XaxAB pore complex with other α-PFTs.
DOI: https://doi.org/10.7554/eLife.38017.021

*figure supplements 7–8*). These residues likely interact with negatively charged lipid head groups of target membranes and thereby stabilize the pore complex in the lipid bilayer.

When comparing the shape of XaxAB with that of the pores of FraC and ClyA, we found that the crown-like structure of XaxA is shared by actinoporin FraC (*Tanaka et al., 2015*) but not by ClyA (*Mueller et al., 2009*), where the extramembrane regions form a cylinder (*Figure 3—figure supplement 9a*). In agreement with the smaller number and size of subunits in FraC and ClyA, these pores have a smaller diameter than the XaxAB pore, and, in addition, FraC contains large β-sheets in the extramembrane region (*Figure 3—figure supplement 9b*). Interestingly, the lumen of all pores is negatively charged (*Figure 3—figure supplement 9c*), suggesting the same preference for positively charged molecules.

## Interaction between XaxA and XaxB in the pore complex

The tail domains of XaxA and XaxB do almost not differ between the oligomeric pore conformation and soluble monomers. The neck and head domains of XaxA are also arranged similarly to the crystal structure, however, the coiled-coil is twisted by 15 Å and interacts with helices αB and αC of the adjacent XaxB (*Figure 4a,c*, *Figure 5*). The neck and head domains of XaxB, however, differ considerably in comparison to the soluble monomer. The amphipathic helix αF and the helix-loop-helix motif fold out, thereby extending helices αD and αG forming the transmembrane region (*Figure 4b, d*, *Figure 5*).

The tail and head domains of XaxA and XaxB mediate interactions between the proteins in the heterodimer. We identified four major interfaces, two in the tail and two in the head domain region. The interfaces between the tail domains are stabilized by several putative hydrogen bonds and electrostatic interactions between helices αG and the C-terminal helix αH of XaxA and helices αB, αC and the C-terminal helix αH of XaxB (*Figure 5a–c*). Dimerization of XaxA and XaxB probably helps stabilizing the tail domain of XaxB, which takes different positions in the crystal structure (*Figure 5a*, *Figure 1—figure supplement 3*).

The first interface between the head domains is formed by helices αD and αG of XaxA that interact with helices αD and αG of XaxB via a putative hydrogen network and salt bridges (*Figure 5d*). The second one is mediated by hydrophobic interactions between helices αF and αE of XaxB with αD and αG of XaxA (*Figure 5f*). A prominent feature is the high accumulation of aromatic residues at this interface (*Figure 5e*). Interestingly, some of these residues are also involved in stabilizing the soluble XaxB monomer (*Figure 2*). Since most of the interfaces between XaxA and XaxB in the heterodimer locate to the tail domain and do not differ between the soluble monomer and pore conformation, we suggest that heterodimer formation precedes membrane insertion.

The heterodimers are linked manifold in the oligomeric pore. One XaxA interacts simultaneously with XaxA and XaxB of the adjacent heterodimer. The same is true for XaxB that interacts with both XaxA and XaxB of the adjacent heterodimer (*Figure 6*).

Two major interfaces are mediated by the tail domains of XaxA and XaxB (*Figure 6a–c*). The residues K45, N50, E398, E402 and D333 that are conserved in XaxA form an extensive putative

**Table 2.** EM data collection and refinement statistics of XaxAB.

| Data collection | |
|---|---|
| Microscope | Titan Krios (Cs corrected, XFEG) |
| Voltage (kV) | 300 |
| Camera | Falcon III (counting mode) |
| Magnification | 59 k |
| Pixel size (Å) | 1.11 |
| Number of frames | 180 |
| Total electron dose (e⁻/Å²) | 44 |
| Exposure time (s) | 60 |
| Defocus range (μm) | 1.0–2.6 |
| Number of particles | 139,286 |
| **Atomic Model Composition** | |
| Non-Hydrogen atoms | 72,436 |
| Protein Residues | 9139 |
| **Refinement (Phenix)** | |
| RMSD bond | 0.006 |
| RMSD angle | 0.98 |
| Model to map fit, CC mask | 0.85 |
| Resolution (FSC@0.143, Å) | 4.0 |
| Map sharpening B-Factor (Å²) | −170 |
| **Validation** | |
| Clashscore, all atoms | 4.68 |
| Poor Rotamers (%) | 0.92 |
| Favored rotamers (%) | 94.56 |
| Ramachandran outliers (%) | 0 |
| Ramachandran favored (%) | 97.42 |
| Molprobity score | 1.35 |

DOI: https://doi.org/10.7554/eLife.38017.022

hydrogen network and salt bridges with helices αB (R48, Y52) and αC (D138, R147) of the adjacent XaxB (*Figure 6b*). A second putative hydrogen network between helices αC2 and αG in XaxA and helices αD and αG in XaxB likely contributes to the stabilization of the oligomer (*Figure 6c*). The oligomer is further stabilized by a putative salt bridge between two XaxAs. Glutamate E206 in the loop between αC1 and αC2 of XaxA of one subunit interacts with lysine K405 in the C-terminal helix of neighboring XaxA (*Figure 6d*). The fourth interface is formed by the head domains of two XaxBs via several putative hydrogen bonds (D197/S241 N192/K245) (*Figure 6e*). Taken together the heterodimeric subunits of the complex and the heterodimer itself are stabilized by strong interactions that guarantee a stable pore complex inside the membrane.

## XaxAB spontaneously inserts membranes

There are at least two concerted or consecutive steps during pore formation of PFTs, namely oligomerization and membrane penetration (*Cosentino et al., 2016*). In bi-component toxins, where both proteins contribute to building the pore, the two components first dimerize into a heterodimer prior to oligomerization (*Colletier et al., 2016*; *Badarau et al., 2015*). In several cases, PFTs have been shown to oligomerize and insert spontaneously into membranes in vitro (*Iacovache et al., 2008*). However, membrane insertion in vivo depends on the specific interaction with lipids or proteins on the membrane surface of the host (*Ros and García-Sáez, 2015*). To better understand the process

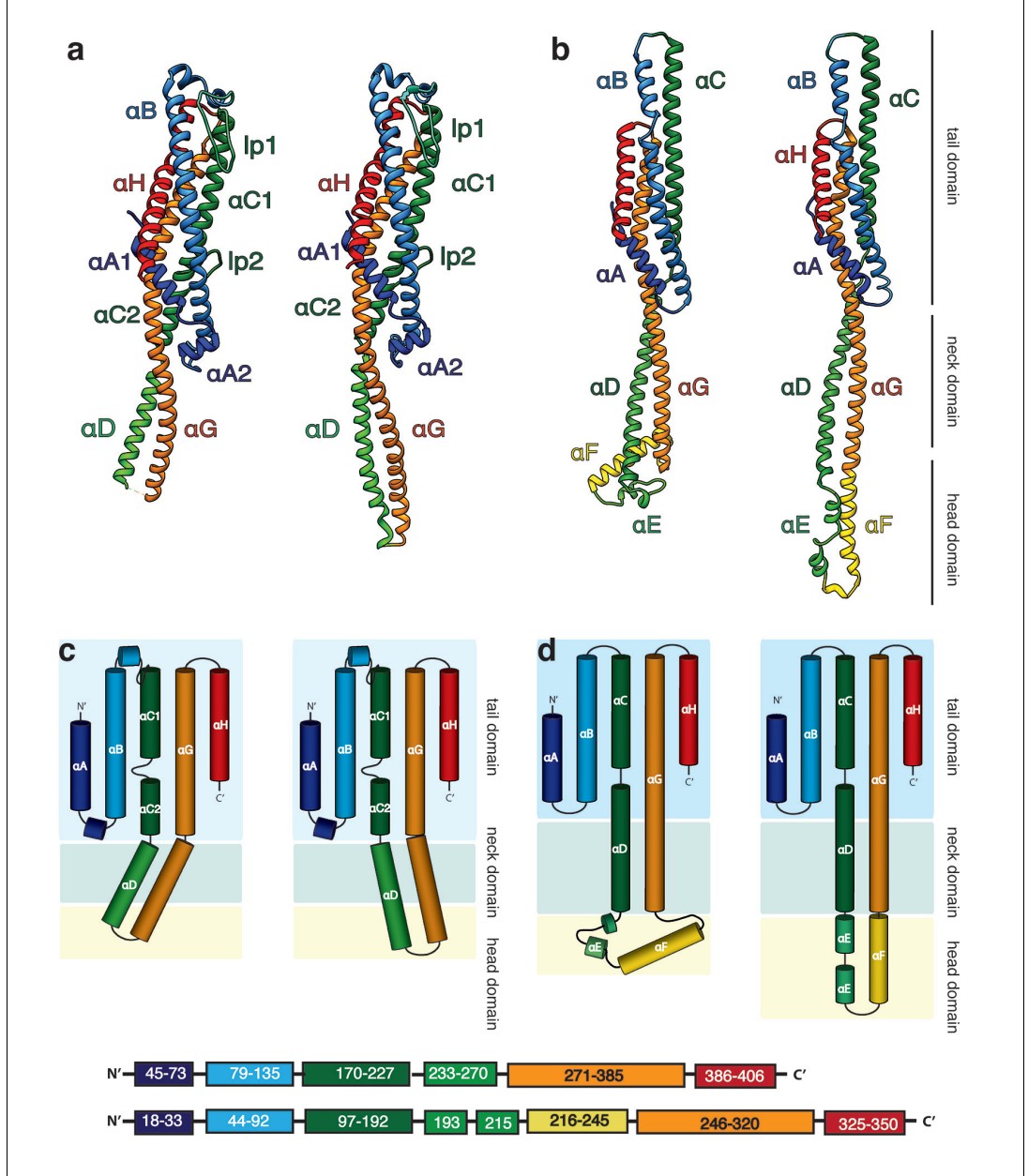

**Figure 4.** Structures of the soluble monomer and protomer of XaxA and XaxB. (a) Ribbon representation of the atomic model of the XaxA monomer (left) and protomer (right). (b) Ribbon representation of the XaxB monomer (left) and protomer (right). (c–d) Topology diagram depicting helices and domain organization of XaxA (c) and XaxB (d). Each helix is shown in a different color and labeled accordingly.
DOI: https://doi.org/10.7554/eLife.38017.024

The following figure supplement is available for figure 4:

**Figure supplement 1.** Comparison of XaxAB and YaxAB.
DOI: https://doi.org/10.7554/eLife.38017.025

of dimerization, membrane insertion and pore formation of XaxA and XaxB, we performed in vitro reconstitution assays with and without liposomes.

XaxA alone has the tendency to form small aggregates by interacting with its head domain (*Figure 1—figure supplement 1d–e*). Since the short hydrophobic region of the head domain resides inside the membrane in the pore complex, we believe that the clustering of XaxA monomers is caused by mild hydrophobic interactions of these regions. This again suggests that already the

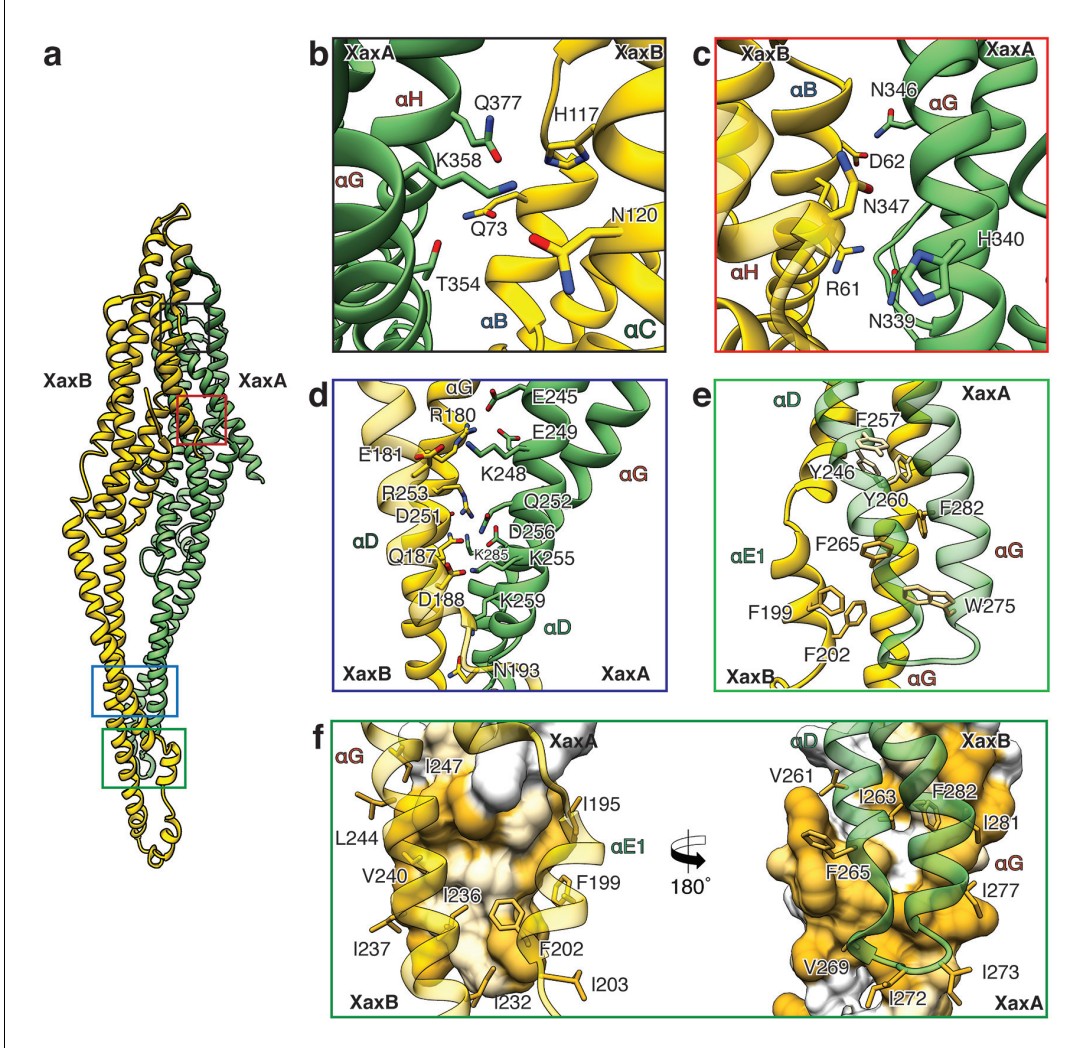

**Figure 5.** XaxAB heterodimer interactions in the pore complex. (**a**) Overview of interaction interfaces between XaxA and XaxB. (**b–c**) Network of putative hydrogen bonds between the tail domains. (**d**) Putative salt bridges in the junction connecting the neck and head domains. (**e**) The hydrophobic head domains of XaxA and XaxB are stabilized by a cluster of aromatic amino acids. (**f**) Hydrophobic interface between the transmembrane domain of XaxA and XaxB in one subunit of the pore. Left: XaxB is depicted in ribbon and XaxA in surface representation colored by hydrophobicity. Right: XaxA is depicted in ribbon representation and XaxB in surface representation colored by hydrophobicity. Protomers of XaxA and XaxB are depicted in green and yellow, respectively. Helices not involved in the interaction are shown in a lighter color for visualization purposes.

DOI: https://doi.org/10.7554/eLife.38017.026

soluble monomeric form of XaxA has a certain affinity to the hydrophobic environment of biomembranes. To test, whether XaxA can spontaneously insert into or associate with membranes, we incubated it with 1-palmitoyl-2-oleoyl-sn-glycero-3-phosphocholine (POPC) or brain polar lipids (BPL) liposomes and analyzed its incorporation by size exclusion chromatography and negative stain electron microscopy (*Figure 7a–d*). Interestingly, the protein was not incorporated into the liposomes and no larger structures could be observed on the vesicles (*Figure 7c,d*). This indicates that albeit the hydrophobic tip of the head domain, XaxA cannot spontaneously insert blank membranes in vitro. The same is true for XaxB alone. When incubated with liposomes the protein neither perforates membranes nor oligomerizes on the liposomes (*Figure 7a,b,e,f*).

When both XaxA and XaxB were added to liposomes, they spontaneously associated with the vesicles and formed the typical crown-shaped structures (*Figure 7g–i*) as we have observed them in detergents (*Figure 3—figure supplement 1c*). Notably, this is independent of the sequence of mixing, that is XaxA can be added before XaxB or vice versa, suggesting that dimerization of XaxA and

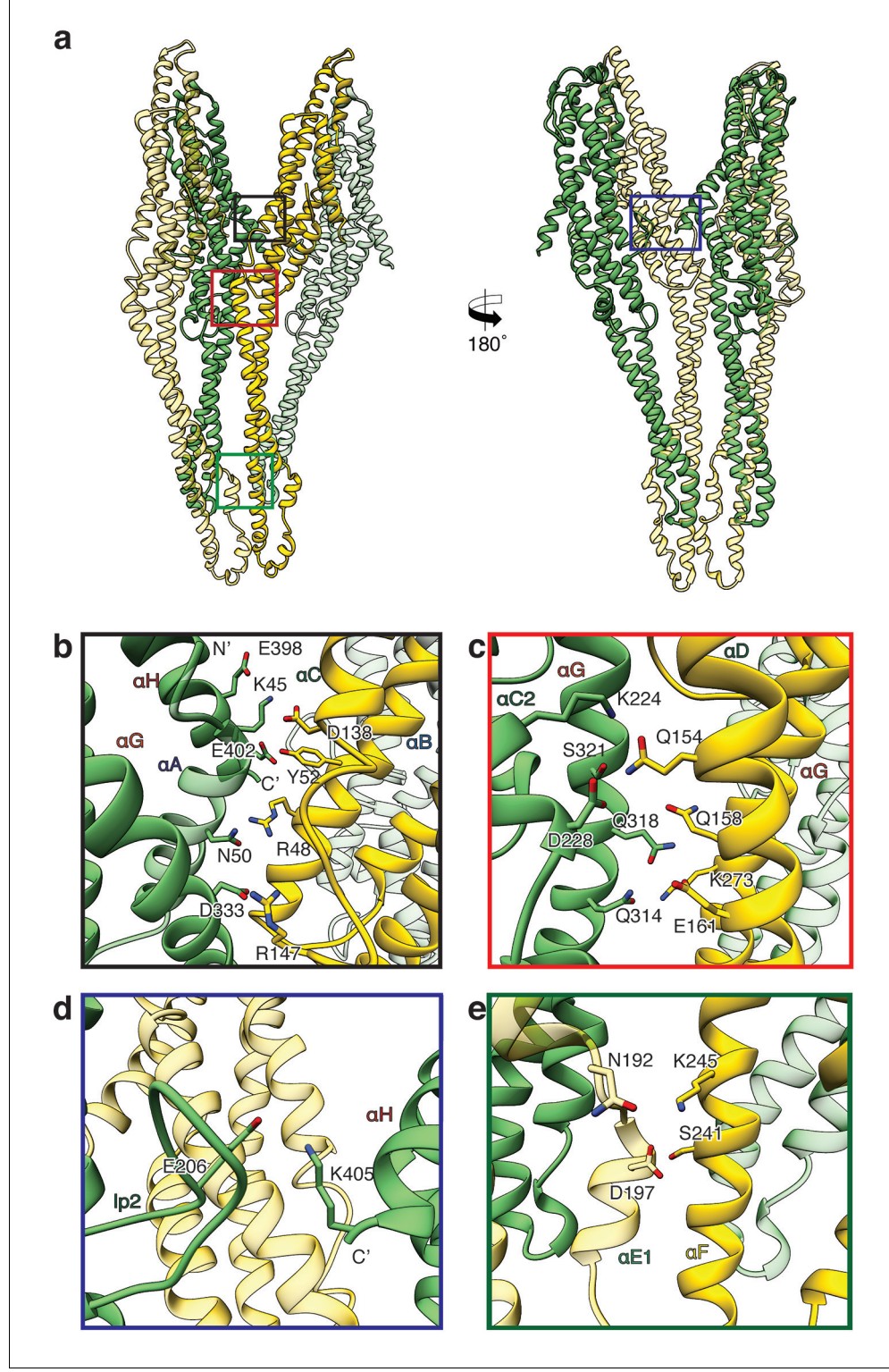

**Figure 6.** Inter-subunit interfaces of the XaxAB pore complex. (**a**) Overview of four prominent inter-subunit interfaces. (**b,c**) The tail domain of XaxA forms an extensive putative hydrogen network with the tail and neck domain of the adjacent XaxB. (**d**) A putative salt bridge formed between the C-terminus and the loop connecting αC1 and αC2 of neighboring XaxA protomers further stabilizes the pore complex. (**e**) Stabilization of the transmembrane pore by additional putative hydrogen bonds and a salt bridge formed between XaxA and XaxB

*Figure 6 continued on next page*

*Figure 6 continued*
from the adjacent subunit. Protomers of XaxA and XaxB are depicted in green and yellow, respectively. Helices not involved in the interaction are shown in a lighter color for visualization purposes.
DOI: https://doi.org/10.7554/eLife.38017.027

XaxB is necessary for spontaneous association of the proteins with the membrane and subsequent pore formation. Importantly, association with liposomes happens without specific lipids, such as cholesterol, or protein receptors at the membrane surface. At this point, we cannot distinguish between association with and insertion into the membranes. Thus, formation of a pre-pore before membrane insertion cannot be excluded.

## Pore formation – structural comparison between monomers and pores

In general, the transition from the soluble monomer to the protomer does not involve major structural rearrangements of the whole molecule. Only the conformation of the head domains changes considerably. Besides the described twist of XaxA (*Figure 4a*), the α-helical tongue αF of XaxB folds out forming the transmembrane region. Interestingly, the conformation of the coiled-coil backbone in XaxB remains unaltered (*Figure 4b*). This is in direct contrast to ClyA (*Wallace et al., 2000*) but similar to FraC (*Tanaka et al., 2015*), the only other two α-PFTs, for which a structure of the soluble and pore complex has been determined at high resolution. In ClyA, not only the head domain but also the tail domain undergoes considerable conformational changes (*Mueller et al., 2009*).

In order to better understand the conformational changes during dimerization, oligomerization and pore formation, we compared the structures of the soluble and pore forms of XaxA and XaxB. When the crystal structures of XaxA and XaxB are overlaid with the respective XaxAB structure, it becomes obvious that the neck and head domains of the proteins would not interact (*Video 2*). In agreement with our reconstitution assays such a dimer would probably not be able to spontaneously insert into membranes. In the XaxAB pore conformation, however, helices αD and αG of XaxA, forming the coiled-coil backbone twist by 15 Å toward XaxB (*Figure 4a*, *Figure 5*, *Video 2*). As described above, through this conformational change a stronger interaction with XaxB is created. Interestingly, without the conformational change in XaxA, oligomerization of XaxAB would not be possible because of prominent steric hindrances (*Video 3*). This movement is therefore crucial for complex formation.

If we assumed that only XaxA and not XaxB changed its conformation during dimerization and oligomerization (*Video 2*, *Video 3*), the transmembrane region of XaxA would sterically clash with the loop between αF and αG of XaxB from the adjacent subunit (*Videos 2* and *3*). This could in principle trigger conformational changes in XaxB that activate its head domain for membrane insertion.

To better analyze these hypothetical conformational changes in detail, we created a heterodimer model comprising the cryo-EM structure of XaxA (XaxA$_{prot}$) and the crystal structure of XaxB (XaxB$_{mon}$) and analyzed interfaces and residues that might trigger membrane insertion (*Figure 8*). We identified two hinge regions that facilitate the swinging out movement of αF in XaxB (*Figure 8*, *Video 2*, *Video 3*). One hinge region is located in the hydrophobic loop between the short helices of the helix-loop-helix motif. It contains a highly conserved proline residue (P204) that is also involved in stabilization of the soluble monomer (*Figure 2*). The second hinge is located in the loop connecting αF and αG, including the conserved residue G243 (*Figure 2*, *Figure 8*).

A cluster of aromatic residues at the bottom of the head domain of our XaxA$_{prot}$-XaxB$_{mon}$ heterodimer model suggests that this region could be crucial in triggering the conformational changes in XaxB when exposed to a lipid membrane. In the heterodimer of the XaxAB pore complex, most of these residues build a hydrophobic cluster between the transmembrane domain of XaxA and the reorganized helix αE of XaxB (*Figure 8*). Aromatic residues have been shown to be important for membrane insertion of many PFTs and responsible for conformational changes induced by their interaction with membranes or detergents (*Mueller et al., 2009*). Interactions with the membrane likely destabilize this region, inducing stronger conformational changes in the rest of the domain.

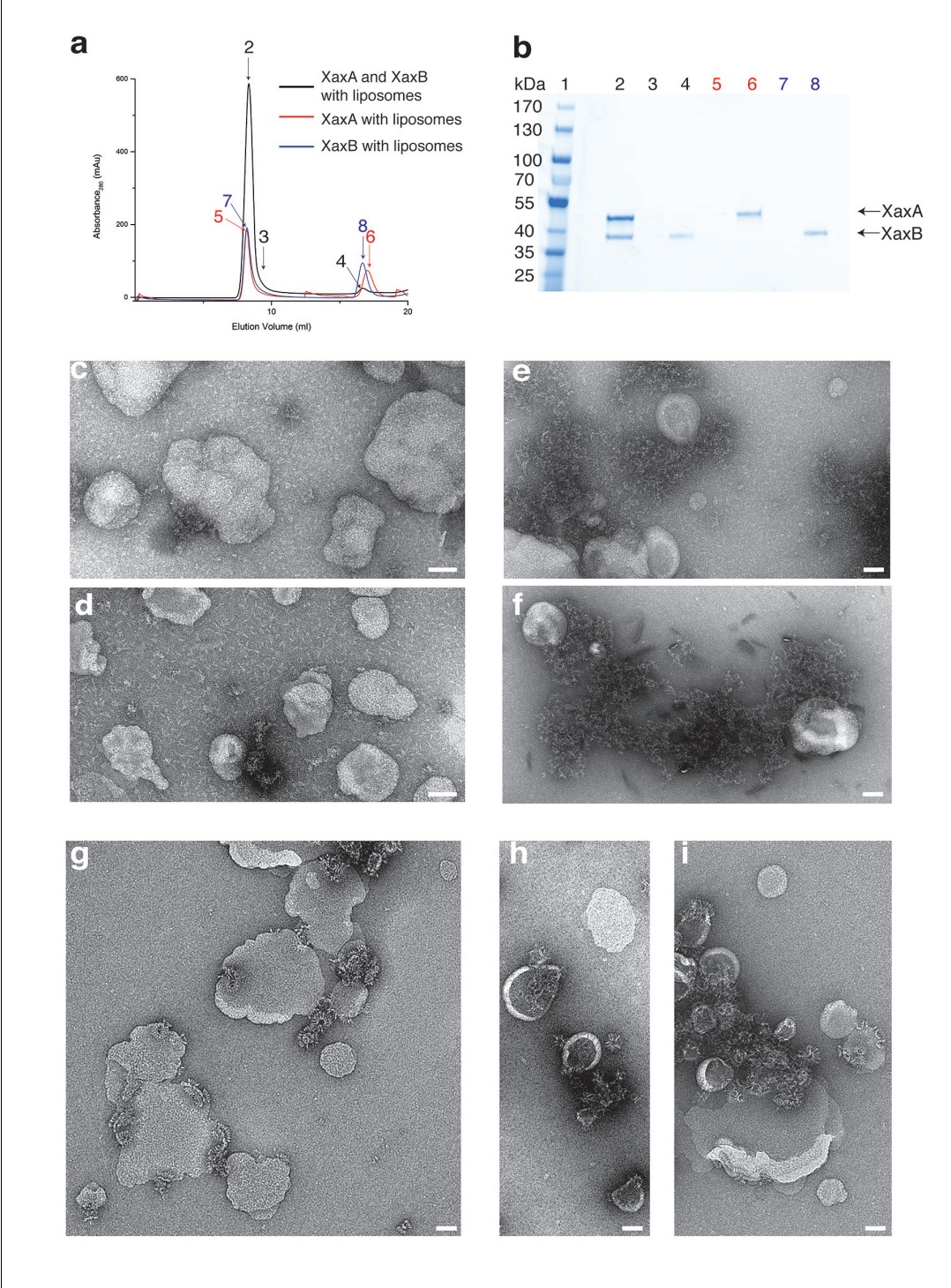

**Figure 7.** XaxAB reconstitution in liposomes. (**a**) Size exclusion profiles of XaxA (red) and XaxB (blue) alone and of a 1:1 mixture of XaxA and XaxB (black) after incubation with liposomes. Arrows and numbers indicate the fractions corresponding to the lanes in the SDS-PAGE gel. (**b**) SDS-PAGE of the peak fractions of (**a**). Lane 1: molecular weight marker, lanes 2–3, 4: void volume and monomeric peak of the XaxA/XaxB mixture, lanes 5–6: void volume and monomer peak of XaxA, lanes 7–8: void volume and monomer peak of XaxB. (**c–d**) Negative stain EM of XaxA reconstitutions into POPC (**c**) or BPL (**d**) liposomes. (**e–f**) Negative stain EM of XaxB reconstitutions into POPC (**e**) or BPL (**f**) liposomes. (**g–i**) Negative stain EM of XaxAB reconstitutions into POPC (**g**) or BPL (**h–i**) liposomes. Scale bars 50 nm.

DOI: https://doi.org/10.7554/eLife.38017.028

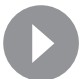

**Video 2.** Dimerization and conformational change of XaxA and XaxB leading to the final pore complex. Starting from the soluble monomers of XaxA and XaxB, the video focuses on the conformational changes during dimerization and membrane insertion.
DOI: https://doi.org/10.7554/eLife.38017.029

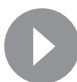

**Video 3.** Interaction between the head domains of XaxA and XaxB contributes to membrane insertion of XaxB. The video highlights possible intermediate interactions and clashes during oligomerization and membrane insertion. It starts with XaxA and XaxB in their monomeric conformation in the position of the respective protomers in the pore. Then shifts to XaxA in its pore conformation, followed by a conformational change in XaxB leading to the final XaxAB in the pore complex. Dimerization of the soluble monomers would introduce a large steric clash between the head domains. Therefore, the soluble monomer of XaxA must transition to its protomeric form prior to oligomerization. The remaining smaller steric clash of XaxA with the loop between helices αD, αG in the head domain of XaxB probably destabilizes its conformation and activates XaxB for membrane insertion.
DOI: https://doi.org/10.7554/eLife.38017.030

## Mechanism of pore formation

Our atomic model of XaxA and XaxB in solution as well as in the pore conformation provides important insights into the interaction and function of these proteins. Although the structural record is lacking intermediate states, we can use the information provided by our structural data to define critical steps in the action of XaxAB toxins and suggest the following mechanism.

Although XaxA and XaxB are not homologous, their structure is very similar. The two components of the xenorhabdolysin form heterodimers, 12 to 15 of which assemble into membrane-perforating pores. In contrast to previous predictions (*Vigneux et al., 2007*; *Zhang et al., 2014*), XaxAB is therefore not a typical binary toxin, but rather a bi-component α-PFT. So far, only structures of bi-component β-PFTs have been reported. Our structure of the XaxAB pore represents the first structure of a bi-component α-PFT.

Our results show that XaxA and XaxB together form higher oligomers in the absence of detergent or membranes. In addition, XaxA likely activates XaxB during oligomerization by inducing conformational changes. We therefore propose that XaxA and XaxB dimerize (*Figure 9a–c*, *Figure 9—figure supplements 1a–c* and *2a–c*) and oligomerize (*Figure 9d*, *Figure 9—figure supplements 1d* and *2d*) in solution. Dimerization happens probably spontaneously since the conformation of domains located at the heterodimer interface in the tail domains of XaxA and XaxB is not different compared to the monomers. The conformational change in the neck and head domain of XaxA (*Figures 9b* and *5d–f*) further stabilizes the interaction and is crucial for oligomerization (*Figure 9d*). During oligomerization XaxA sterically clashes with the loop connecting helices αF and αG in XaxB. We therefore propose that XaxA induces conformational changes in XaxB that do not immediately result in exposing its hydrophobic domain but rather put XaxB in an activated state for membrane insertion (*Figure 9b–c*). When interacting with a lipid membrane, aromatic residues at the bottom of the head domain of XaxB likely trigger the conformational change resulting in membrane perforation (*Figure 9d*). Our reconstitution assays in liposomes showed that neither XaxA nor XaxB strongly interact with liposomes. Thus, neither the interaction of the aromatic residues of XaxB nor the hydrophobic domain of XaxA are able to enter membranes on their own and dimerization and induced conformational changes during oligomerization are crucial for membrane insertion. Since XaxB is the component that finally forms the pore, XaxA that only partially enters

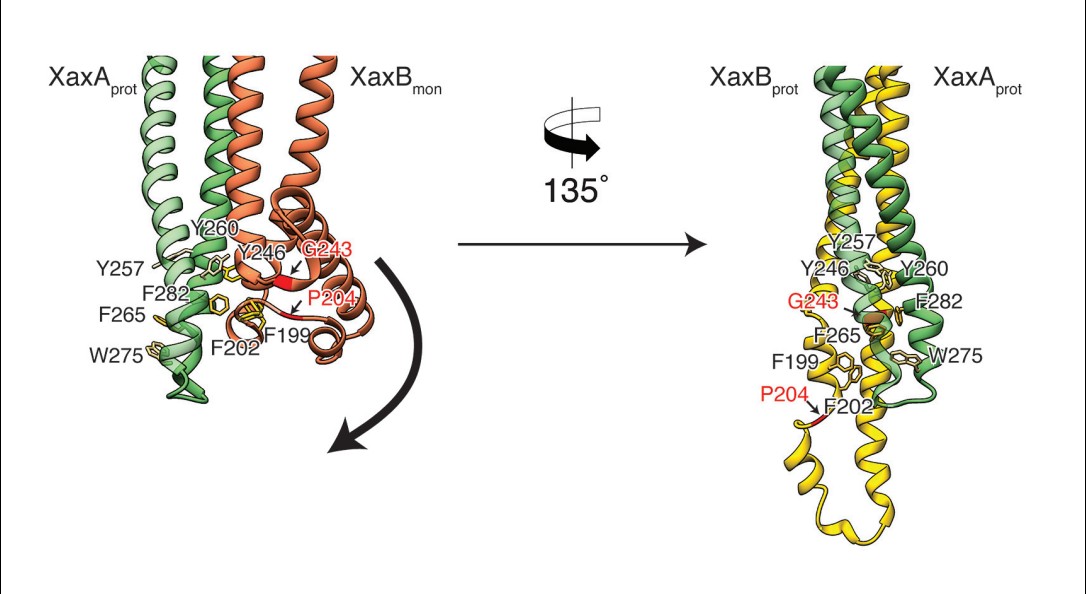

**Figure 8.** Model for membrane insertion. A heterodimer model was built with XaxA in protomeric (XaxA_{prot}) and XaxB in monomeric (XaxB_{mon}) conformation to mimic a possible intermediate state (left) and compared to the conformation in the pore complex (right). An aromatic cluster at the bottom of the head domain of the XaxA_{prot}-XaxB_{mon} heterodimer possibly triggers the conformational change of XaxB when exposed to a lipid membrane. Swinging out of the amphipathic helix αE happens at two hinge regions at the position of conserved proline (P204) and glycine (G243) residues, respectively (highlighted in red and marked with arrows). After membrane insertion, the aromatic residues interact with each other, stabilizing the new conformation. The soluble monomer of XaxB is shown in orange. Protomers of XaxA and XaxB are depicted in green and yellow, respectively.
DOI: https://doi.org/10.7554/eLife.38017.031

the membrane, acts like an activator of XaxB and stabilizes it in the pore complex.

Recently, a new mechanism for ClyA membrane permeation has been suggested in which a conformational change in a ClyA monomer initiates the assembly of dimers and higher oligomers on the membrane forming a homo-dodecameric pre-pore complex that ultimately enters the membrane after an additional conformational change (*Benke et al., 2015*). Although, we never observed structures at high resolution that would indicate a pre-pore complex, we cannot exclude that such a complex exists as intermediate state on liposomes before permeation (*Figure 9e*, *Figure 9—figure supplements 1e* and *2e*). Obviously, more evidence is needed before our proposed mechanism of XaxAB action can be regarded as established. Thus, additional structures of intermediate states are needed to fully understand the process.

In summary, our results provide novel insights into the mechanism of action of xenorhabdolyins and serve as a strong foundation for further biochemical experiments to fully understand the molecular mechanism of xenorhadolysin intoxication.

## Comparison to YaxAB

During the revision of our work, crystal structures and a low-resolution cryo-EM structure of the human pathogenic homolog YaxAB from *Yersinia enterocolitica* as well as a crystal structure of PaxB from *Photorhabdus luminescens* have been published (*Bräuning et al., 2018*). The crystal structures of YaxA and XaxA, as well as YaxB (PaxB) and XaxB are very similar although their sequences are only 54.5 and 36.0% (56.7%) identical, respectively (*Figure 4—figure supplement 1a,c,d*). Importantly, YaxA does not contain the hook-shaped loop (lp1), which is a prominent feature of XaxA (*Figure 4—figure supplement 1a,b*). The neck and head domain of XaxA and YaxA as well as the head of XaxB and PaxB differ in their position indicating that these domains are flexible in solution. This is supported by the fact that the head domain of XaxA and YaxB are not resolved in the structures. Similarly, the tip of the tail domain takes different positions in XaxB and is not resolved in YaxB (*Figure 4—figure supplement 1c,d*).

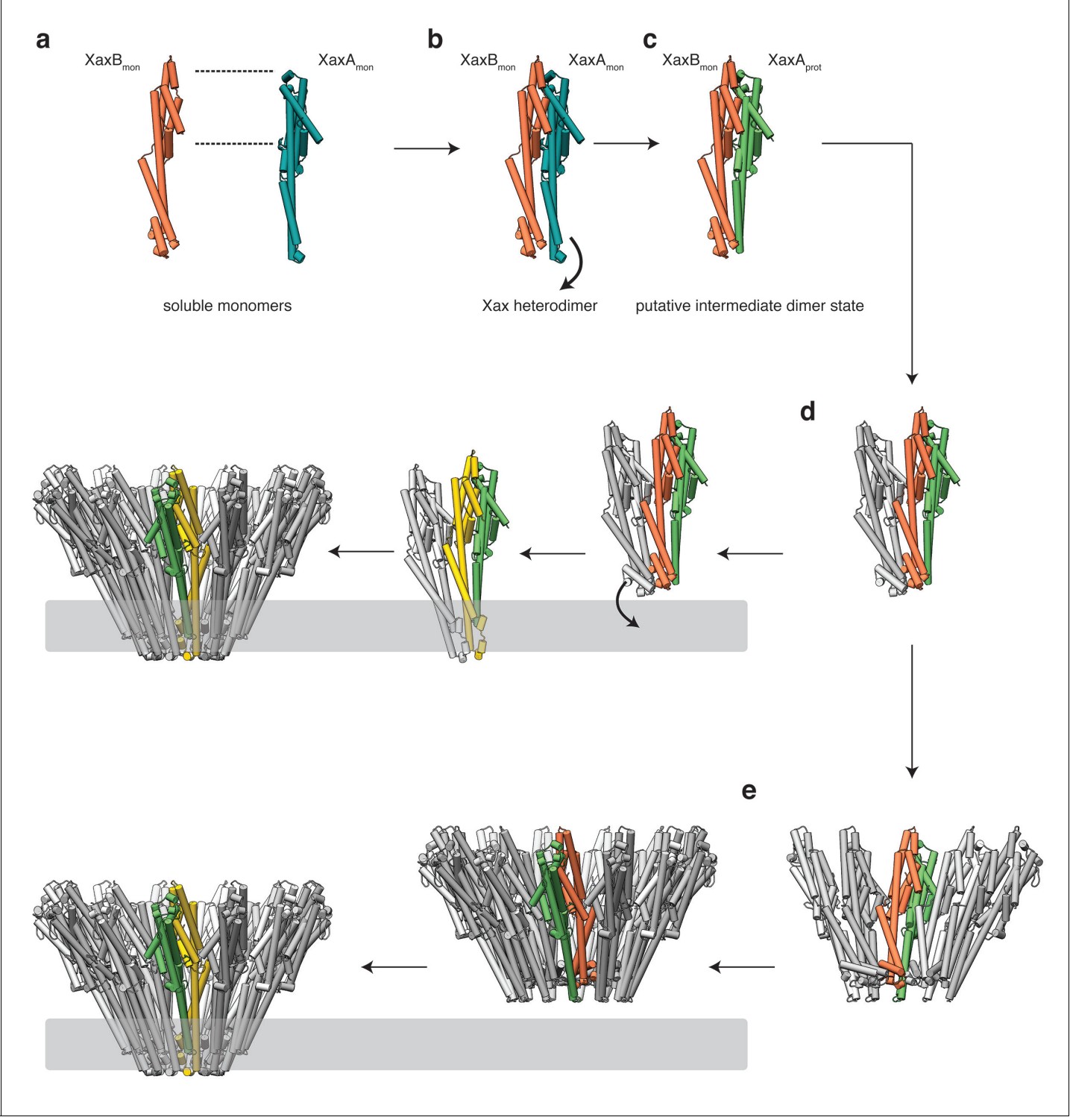

**Figure 9.** Mechanism of pore formation. (a) XaxA and XaxB dimerize in solution. (b–c) The major interaction site in the heterodimer is between the tail domains of XaxA and XaxB (b). This interaction induces neck and head domain (αD, αG) of XaxA to shift towards XaxB (αD, αG) activating XaxB for oligomerization interaction (c) and membrane insertion by clashing with the loop between αE and αF. (d–e) Interactions of aromatic residues at the bottom of the head domain with the membrane trigger the conformational changes that lead to membrane insertion. Membrane insertion happens either directly (d) or after a pre-pore complex is formed (e). The soluble monomer of XaxA and XaxB is shown in teal and orange, respectively. After the conformational change of the soluble monomers, XaxA and XaxB protomers are depicted in green and yellow, respectively.

DOI: https://doi.org/10.7554/eLife.38017.032

*Figure 9 continued on next page*

*Figure 9 continued*

The following figure supplements are available for figure 9:

**Figure supplement 1.** Mechanism of pore formation.
DOI: https://doi.org/10.7554/eLife.38017.033

**Figure supplement 2.** Mechanism of pore formation.
DOI: https://doi.org/10.7554/eLife.38017.034

Although the authors used the same detergent for pore assembly and also stabilized the pores in amphipols, the YaxAB pore complex comprises 8 to 12 heterodimers in contrast to the 12 to 15 heterodimers in our XaxAB pore. This suggests a species-dependent size variability. The protomer structures of YaxAB and XaxAB are very similar (RMSD of 1.145 and 1.252, respectively). Significant differences, however, can be seen in the head domains which could in principle indicate structural differences between the proteins (*Figure 4—figure supplement 1b,e*). However, since the relatively low resolution of the YaxAB pore structure (5.5 Å) impedes an accurate building of the atomic model, only a high-resolution structure of YaxAB would enable a proper comparison.

Interestingly, YaxA associates directly with erythrocyte membranes. This is in direct contrast to our findings showing that XaxA does interact with liposomes in vitro. This results in different models. Whereas Bräuning et al. hypothesize that YaxA enters the membrane first and then recruits YaxB, we propose that XaxA and XaxB already heterodimerize/oligomerize in solution and then associate with the membrane as heterodimers or oligomers.

## Materials and methods

**Key resources table**

| Reagent type (species) or resource | Designation | Source or reference | Identifiers | Additional information |
|---|---|---|---|---|
| Gene (*Xenorhabdus nematophila*) | XaxA | N/A | NCBI Reference sequence: FN667742.1 | Genes ordered from GenScript |
| Gene (*X. nematophila*) | XaxB | N/A | NCBI Reference sequence: FN667742.1 | Genes ordered from GenScript |
| Cell line (*Escherichia coli*) | BL21-CodonPlus (DE3)-RIPL | Agilent Technologies | Agilent: 230280–41 | |
| Recombinant DNA reagent | pET19b | Novagen | Merck: 69677 | |
| Chemical compound, drug | Cymal-6 | Anatrace | Anatrace: 228579-27-9 | |
| Chemical compound, drug | Amphipol A8-35 | Anatrace | Anatrace: 1423685-21-5 | |
| Software, algorithm | SPHIRE software package | *Moriya et al. (2017)* PMID: 28570515 | | |
| Software, algorithm | Gautomatch | N/A | | http://www.mrc-lmb.cam.ac.uk/kzhang/ |
| Software, algorithm | Phenix | *Terwilliger et al. (2008)* PMID: 18094468 | | |
| Software, algorithm | UCSF Chimera | *Pettersen et al. (2004)* PMID: 15264254 | | |
| Software, algorithm | hkl2map | *Pape and Schneider, 2004* ISSN: 0021–8890 | | |
| Software, algorithm | Crank2 | *Pannu et al. (2011)* PMID: 21460451 | | |

### Protein expression and purification

The genes coding for C-terminally His$_6$-tagged XaxA and N-terminally His$_6$-tagged XaxB were introduced into a pET19b vector and expressed in the *E. coli* BL21 RIPL (DE3) expression strain. Both constructs contained a PreScission cleavage site. For the expression culture, 2 l of LB media containing 125 µg/ml ampicillin were inoculated with the preculture and cells were grown at 37°C until an OD$_{600}$ of 0.5–0.8 was reached. Selenomethionine-substituted XaxB was expressed in the *E. coli* BL21

RIPL (DE3) strain in M9 minimal medium with the addition of 100 mg/l L-lysine, 100 mg/l L-phenylalanine, 100 mg/l L-threonin, 50 mg/l L-isoleucine, 50 mg/l L-leucine 50 mg/l L-valine and finally 60 mg/l l-selenomethionine (SeMet). Afterwards, protein production was induced by adding 0.4 mM of isopropyl-β-D-thiogalactopyranoside (IPTG) and incubated for 20 hr at 20°C. The cells were harvested and the bacterial pellet homogenized in a buffer containing 50 mM HEPES, pH 7.5, and 200 mM NaCl. After cell disruption, the lysate was centrifuged at 38,000 rpm, 4°C and XaxA and XaxB was purified using Ni-NTA affinity and size-exclusion chromatography (Superdex 200 26/600, GE Healthcare).

## Crystallization of XaxA and XaxB

Crystallization experiments were performed using the sitting-drop vapor diffusion method at 20°C. XaxA crystals formed by mixing 0.1 µl of 40 mg/ml purified XaxA with 0.1 µl reservoir solution containing 0.2 M sodium chloride, 0.1 M phosphate citrate pH 4.2% and 10% PEG 3000 over a period of 3 weeks. SeMet-labeled XaxB (40 mg/ml) was mixed in a 1:1 ratio with reservoir solution containing 0.2 M NaBr, 0.1 KCl and 20% PEG 3350 with a final drop size of 2 µl. Prior to flash freezing in liquid nitrogen, the crystals were soaked in reservoir solution containing 20% glycerol as cryoprotectant.

## X-ray data collection and processing

X-ray diffraction data for XaxA was collected at the PXIII-X06DA beamline at the Swiss Light Source (24 datasets) and at the DESY PETRA III beamline P11 (3 datasets) from one crystal. The datasets were merged and used for phase determination.

Data collection for XaxB was performed at the PXII-X10SA beamline. Datasets were indexed, integrated and merged with the XDS package (*Kabsch, 2010b*, *2010a*).

## Structure solution and refinement

XaxA crystallized in orthorhombic space group $P2_12_12_1$ with a unit cell dimension of $67 \times 90 \times 153$ Å and two molecules per asymmetric unit (AU). Phases were determined using the anomalous sulfur signal of the merged datasets and HKL2MAP (*Pape and Schneider, 2004*), the graphical interface for SHELX C/D/E (*Sheldrick, 2010*). The obtained phases combined with the given sequence and a few placed α-helices in the density with COOT (*Emsley et al., 2010*) were sufficient enough for phenix autobuild (*Terwilliger et al., 2008*) to almost completely build the structure of XaxA. The structure was refined with the datasets collected at the DESY PETRA III beamline P11. XaxB also crystallized in orthorhombic space group $P2_12_12_1$ with a unit cell dimension of $89 \times 99 \times 194$ Å and four molecules per AU. The diffraction data of XaxB was processed with the XDS package and SeMet atoms were determined using the CRANK2 pipeline (*Ness et al., 2004*; *Pannu et al., 2011*) in the CCP4 software package. SHELX C/D (*Winn et al., 2011*) was used in the substructure detection process, while REFMAC (*Skubák et al., 2004*), SOLOMON and PARROT (*Abrahams and Leslie, 1996*) were used for phasing and substructure refinement and density modification for hand determination, respectively. BUCANEER (*Cowtan, 2012*) gave the best results for the initial model-building step. This model was first optimized with phenix autobuild (*Terwilliger et al., 2008*). The rest of the model was built in COOT (*Emsley et al., 2010*) using the anomalous peaks of the SeMet residues to determine the amino acid sequence due to the limited resolution. The structures were optimized by iteration of manual and automatic refinement using COOT (*Emsley et al., 2010*) and phenix refine implemented in the PHENIX package (*Adams et al., 2010*) to a final Rfree of 28 and 30% for XaxA and XaxB, respectively (*Table 1*).

## Reconstitution into liposomes

Stock solutions of 10 mg/ml 1-palmitoyl-2-oleoyl-sn-glycero-3-phosphocholine (POPC) and brain extract polar lipids (BPL) (Avanti Polar Lipids) were prepared in buffer containing 20 mM Tris-HCl pH 8, 250 mM NaCl and 5% w/v n-octyl-β-D-glucopyranoside (Antrace). 10 µM XaxA and XaxB were mixed with a final lipid concentration of 2 mg/ml and incubated for 30 min at room temperature. For reconstitution, the mixture was dialyzed against a buffer containing 20 mM HEPES pH 7.5 and 200 mM NaCl. The sample was then analyzed by size exclusion chromatography with a Superose 6 10/300 GL column (GE Healthcare Life Sciences) and by negative stain electron microscopy.

## Preparation of XaxAB pore complexes

XaxAB pore complexes were prepared by incubating equimolar concentrations of XaxA and XaxB with 0.1% cymal-6 (Antrace) at room temperature overnight. For a more homogenous and stable distribution of XaxAB pore complexes, the detergent was exchanged to amphipols A8-35 (Antrace). Amphipols were added in fivefold molar excess and the solution was incubated at room temperature for 20 min. For detergent removal, the mixture was dialyzed against a buffer containing 20 mM HEPES pH 7.5, 200 mM NaCl overnight at room temperature. Subsequently, aggregates and XaxAB pore complexes with higher molecular weight were separated by size exclusion chromatography on a Superose 6 10/300 GL column (GE Healthcare Life Sciences).

## EM data acquisition

The quality of the XaxAB pore complexes was evaluated by negative stain electron microscopy before proceeding to cryo-EM grid preparation. 4 µl of a 0.01 mg/ml XaxAB solution in amphipols were applied to a freshly glow-discharged copper grid (Agar Scientific; G400C) coated with a thin carbon layer and incubated for 45 s. After sample incubation, the solution was blotted with What-man no. four filter paper and stained with 0.75% uranyl formate. The digital micrographs were acquired with a JEOL JEM-1400 TEM equipped with an acceleration voltage of 120 kV, and a $4000 \times 4000$ CMOS detector F416 (TVIPS) with a pixel size of 1.33 Å/pixel.

For sample vitrification, XaxAB pore complexes were concentrated to a final concentration of 1 mg/ml and 4 µl sample was applied onto freshly glow-discharged holey carbon grids (C-flat 2/1, Protochips), incubated for 45 s, blotted for 2.5 s and plunged into liquid ethane with a CryoPlunge3 (Cp3, Gatan) at 90% humidity. The grids were then stored in liquid nitrogen.

A cryo-EM dataset of XaxAB in amphipols was collected with a $C_s$-corrected TITAN KRIOS electron microscope (FEI), with a XFEG and operated at an acceleration voltage of 300 kV. Images were acquired automatically using EPU (FEI) and a Falcon III (FEI) direct detector operated in counting mode at a nominal magnification of 59,000 x corresponding to a pixel size of 1.11 Å/pixel on the specimen level. In total 4746 images were collected with 180 frames, an exposure time of 60 s resulting in a total dose of ~44 $e^-$ $Å^{-2}$ and a defocus range of 1.0–2.6 µm. Motion correction was performed using the MotionCor2 program (*Zheng et al., 2017*).

## Single particle cryo-EM data processing

All image-processing steps were carried out with the SPHIRE software package (*Moriya et al., 2017*) (*Figure 3—figure supplement 4*). Initially, micrographs were manually screened for bad ice or high drift and discarded accordingly. The remaining 3617 motion-corrected sums without dose weighting were evaluated in aspect of defocus and astigmatism in CTER (*Moriya et al., 2017*) and low-quality images were discarded using the graphical CTF assessment tool in SPHIRE (*Moriya et al., 2017*). 186,700 single particles were automatically picked from motion-corrected sums with dose weighting using gautomatch (http://www.mrc-lmb.cam.ac.uk/kzhang/). 2-D class averages were generated as a template for gautomatch by manually picking 200 micrographs with EMAN2 boxer (*Tang et al., 2007*). Pre-cleaning of the dataset and reference-free 2-D classification were performed with the iterative stable alignment and clustering approach ISAC2 (*Yang et al., 2012*) in SPHIRE with a pixel size of 4.97 Å/pixel on the particle level. Refined and sharpened 2-D class averages with the original pixel size and exhibiting high-resolution features were generated with the Beautifier tool implemented in SPHIRE (*Figure 3—figure supplements 3* and *5b*). The quality of the 2-D class averages were examined in regard of high-resolution features and completeness of the XaxAB pore complexes. According to observed oligomerization states of XaxAB pore complexes in the class averages, five initial 3-D models with c12, c13, c14, c15 and c16 symmetry were generated with RVIPER. Particles were then sorted against the five RVIPER models using the 3-D-mulrireference projection matching approach (sxmref_ali3d). The clean dataset was split into four datasets according to the number of XaxAB subunits in the complex: c12: 4409 particles, c13: 53,546 particles, c14: 46,596 particles and c15 34,542 particles. The sixteen-fold symmetry was discarded due to low number of particles (193). The subsets containing particles with 13-, 14- and 15-fold symmetry were further cleaned with ISAC and subsequently subjected to 3-D refinements in MERIDIEN with a mask excluding amphipols and applying c12, c13-, c14-, and c15-symmetry,

respectively (*Moriya et al., 2017*). In the following only the results of the map with the highest resolution will be described in detail.

SPHIRE's PostRefiner tool was used to combine the half-maps, to apply a tight adaptive mask and a B factor of $-170$ Å$^2$. The estimated average resolution according to the gold standard FSC@0.5/0.143 criterion between the two masked half-maps was 4.5/4 Å for the c13-symmetry (*Figure 3—figure supplement 5f*). The estimated accuracy of angles and shifts at the final iteration of the 3-D refinement was 0.55 degrees and 0.6 pixels, respectively. The 'Local Resolution' tool in SPHIRE (*Figure 3—figure supplement 5e*) was used to calculate and analyze the local resolution of the c13 density map. The resulting colored density map showed a local resolution of up to 3.4 Å at the lower tail domain region, whereas the tip of the spikes at the top of the XaxAB pore and at the end of the transmembrane region showed the lowest resolution (5–6.7 Å) (*Figure 3—figure supplement 5e*). The final density was locally filtered according to the estimated local resolution using the 'LocalFilter' tool in SPHIRE. Details related to data processing are summarized in *Table 2*.

## Model building, refinement and validation

The atomic model of the XaxAB pore complex was built by isolating the EM density of a XaxAB dimer and rigid body fitting the crystal structure of XaxA into the EM density map using UCSF Chimera (*Pettersen et al., 2004*). XaxA was further fitted into the dimer density using IMODFIT (*Lopéz-Blanco and Chacón, 2013*). For XaxA only the transmembrane region (aa 254–283) had to be manually built, which was missing in the crystal structure. The final model of the XaxA protomer covers residues 41–405 of the full-length sequence with residues 1–40 missing at the N-terminal helix αA. XaxB was built by placing helix fragments into the remaining density with COOT (*Emsley et al., 2010*), generating first a polyalanine model and subsequently determining the correct sequence by the identification of bulky side chains. The full sequence of the XaxB protomer is also almost covered in the final model (aa 13–350) with the first 12 residues missing at the N-terminal helix αA. The XaxAB dimer was then rigid-body fitted into the XaxAB pore complex using UCSF Chimera (*Pettersen et al., 2004*) and the full model refined using PHENIX real-space refinement (*Adams et al., 2010*). Finally, the overall geometry of the refined model was evaluated with MOL-PROBITY (*Williams et al., 2018*). The data statistics are summarized in *Table 2*.

## Acknowledgements

The crystallographic experiments were performed on the X10SA and X06DA beamlines at the Swiss Light Source, Paul Scherrer Institut, Villigen, Switzerland and beamline P11 at PETRA III, DESY, Hamburg, Germany. We thank the X-ray community at the Max Planck Institute Dortmund and Eckhard Hofmann from the Ruhr-University Bochum for help with data collection and Vincent Olieric and Saravanan Panneerselvam for technical support and help during SAD data collection. We thank T Wagner and C Gatsogiannis for support in electron microscopy image processing and K Vogel-Bachmayr for technical assistance. This work was supported by the Max Planck Society (to SR); the European Council under the European Union's Seventh Framework Programme (FP7/2007–2013) (Grant 615984 to SR)

## Additional information

### Funding

| Funder | Grant reference number | Author |
|---|---|---|
| Max-Planck-Gesellschaft | Open-access funding | Stefan Raunser |
| European Commission | Seventh Framework Programme: ERC 615984 | Stefan Raunser |

The funders had no role in study design, data collection and interpretation, or the decision to submit the work for publication.

## Author contributions

Evelyn Schubert, Data curation, Formal analysis, Validation, Investigation, Visualization, Writing—original draft; Ingrid R Vetter, Data curation, Formal analysis, Validation; Daniel Prumbaum, Data curation; Pawel A Penczek, Software, Methodology; Stefan Raunser, Conceptualization, Supervision, Funding acquisition, Investigation, Writing—original draft, Project administration, Writing—review and editing

## Author ORCIDs

Evelyn Schubert http://orcid.org/0000-0003-2625-1505
Stefan Raunser http://orcid.org/0000-0001-9373-3016

## Decision letter and Author response

Decision letter https://doi.org/10.7554/eLife.38017.043
Author response https://doi.org/10.7554/eLife.38017.044

## Additional files

### Supplementary files

• Transparent reporting form
DOI: https://doi.org/10.7554/eLife.38017.035

### Data availability

The electron density map after post-processing has been deposited to the EMDB under accession code EMD-0088. The final model of XaxAB was submitted to the PDB under the accession code 6GY6. Coordinates of XaxA and XaxB in the soluble form have been deposited in the Protein Data Bank under accession codes PDB 6GY8 and PDB 6GY7.

The following datasets were generated:

| Author(s) | Year | Dataset title | Dataset URL | Database, license, and accessibility information |
|---|---|---|---|---|
| Raunser S, Schubert E | 2018 | Crystal structure of XaxA from Xenorhabdus nematophila | https://www.rcsb.org/structure/6GY8 | Publicly available at the RCSB Protein Data Bank (accession no: 6GY8) |
| Raunser S, Schubert E | 2018 | XaxAB pore complex from Xenorhabdus nematophila | https://www.rcsb.org/structure/6GY6 | Publicly available at the RCSB Protein Data Bank (accession no: 6GY6) |
| Raunser S, Schubert E | 2018 | Crystal structure of XaxB from Xenorhabdus nematophila | https://www.rcsb.org/structure/6GY7 | Publicly available at the RCSB Protein Data Bank (accession no: 6GY7) |

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
