## [Decision Letter]

Thank you for submitting your article "Membrane insertion of α-xenorhabdolysin in near-atomic detail" for consideration by *eLife*. Your article has been reviewed by Richard Aldrich as the Senior Editor, a Reviewing Editor, and three reviewers. The reviewers have opted to remain anonymous.

The reviewers have discussed the reviews with one another and the Reviewing Editor has drafted this decision to help you prepare a revised submission.

Summary:

In the manuscript, the authors describe comprehensive structural studies of pore forming toxin (PFT) XaxAB. The crystal structures of the monomeric forms of XaxA (at 2.5 Å) and XaxB (at 3.4 Å) together with the cryo-EM structure of the assembled heteromeric pore (at 4 Å) provide mechanistic insights into the pore formation by these toxins. The pore complex structure in amphipol shows that the number of the heterodimers varies between 12 and 15 units. The highest resolved pore complex map was the one with 13 heterodimers why the authors fit in the model into this map. Extensive hydrogen network and ionic interactions between XaxA/XaxB heterodimers were identified by the analysis of the pore complex structure. Finally, a model of membrane insertion and pore formation was postulated on the basis of the X-ray structures of the monomers and cryo-EM structure of the pore complex. This is a well-constructed manuscript and body of work that provides new insight into the various mechanisms proteins can use for assembly and pore formation. The authors have done an excellent job of presenting their findings in a way that can be easily appreciated by the reader. The reviewers have appreciated the quality of the work and have focused primarily on the editorial aspects.

Essential revisions:

1) Recently, in parallel with the submission of this manuscript to *eLife*, a manuscript about the structure and function of a similar, a two-component α-PFT (YaxA/YaxB from *Yersinia enterocolitica*) was published (Bräuning et al., 2018). The authors must extensively compare and discuss the above-mentioned manuscript in their revised manuscript.

2) Overall, the work is not sufficiently put into the context of what is known about similar toxin systems to make it of significant interest to broader readership, even as some structural comparison is given. More extensive Introduction and Discussion sections focused on the diversity of toxins and their evolutionary relationships would help to broaden the perspective provided by the articles.

3) The authors make a point of suggesting that XaxA and XaxB can dimerize in solution but with no direct evidence. They should discuss their results more clearly or provide additional experiments to support the point.

4) From the "membrane-inserting" experiments (subsection “XaxAB spontaneously inserts membranes”, Figure 7) whereby liposomes are incubated with XaxA and XaxB, it is not very clear whether the circular-oligomeric XaxAB heterodimers form a pore, as the structures on the liposomes could also represent a pre-form of the pore embedded in the membrane. This needs clarification.

---

## [Author Response]

Essential revisions:1) Recently, in parallel with the submission of this manuscript to eLife, a manuscript about the structure and function of a similar, a two-component α-PFT (YaxA/YaxB from Yersinia enterocolitica) was published (Bräuning et al., 2018). The authors must extensively compare and discuss the above-mentioned manuscript in their revised manuscript.

We now discuss the aforementioned manuscript that has been published after submission of our work in the revised manuscript as suggested.

2) Overall, the work is not sufficiently put into the context of what is known about similar toxin systems to make it of significant interest to broader readership, even as some structural comparison is given. More extensive Introduction and Discussion sections focused on the diversity of toxins and their evolutionary relationships would help to broaden the perspective provided by the articles.

As suggested by the reviewers, we have extended the Introduction to include more details about α-PFTs and β-PFTs in general. We have also included an evolutionary perspective on similar toxins, e.g. ClyA-type toxins. In this context we also discuss now in subsection “Structure of XaxA and XaxB soluble monomers” the loss of function of the rudimentary N-terminal helix observed in the multi-component ClyA-type toxin systems and the new role of the head domain in XaxA and XaxB.

3) The authors make a point of suggesting that XaxA and XaxB can dimerize in solution but with no direct evidence. They should discuss their results more clearly or provide additional experiments to support the point.

In Figure 1—figure supplement 3B, we show by negative stain EM that XaxA and XaxB form oligomeric structures in the absence of detergents. The size and shape of the particles range from dimers to the crown-shaped oligomeric structures resembling partly assembled pores. This is not the case when XaxA or XaxB are imaged separately (see new Figure 1—figure supplement 1C, 2C). We therefore conclude that XaxA and XaxB form heterodimers in solution, which are the building blocks of XaxAB oligomers. However, we are not able to biochemically separate dimers from oligomers. We discuss the results more clearly in the revised manuscript also highlighting the possibility of pre-pore complex formation.

4) From the "membrane-inserting" experiments (subsection “XaxAB spontaneously inserts membranes”, Figure 7) whereby liposomes are incubated with XaxA and XaxB, it is not very clear whether the circular-oligomeric XaxAB heterodimers form a pore, as the structures on the liposomes could also represent a pre-form of the pore embedded in the membrane. This needs clarification.

This reviewer is right. We cannot distinguish if the circular oligomeric XaxAB complexes only associate with the membrane as pre-pore complexes or if they actually insert into the membrane. We describe this now more carefully in the revised manuscript. However, since XaxAB clearly forms pores in detergents and amphipols, we indeed believe that the complexes insert into the membrane before they are completely oligomerized. However, this has to be proven empirically in future experiments. We have discussed this aspect in the revised manuscript and added an alternative route for membrane insertion including a pre-pore complex in our model (Figure 9).